# Analysis of morphological, morphokinetics, cell-free DNA, microRNAs parameters to predict aneuploidy status of embryos

Achmad Kemal Harzif[1,2,3]*, Intan Indah Permatasari[2], Patricia Alika Kurniawan[2], Thalia Amila Elsiyana[2], Fistyanisa Elya Charilda[2], Azizah Fitriayu Andyra[2], Heidi Dewi Mutia[2], Nafi'atul Ummah[1], Putri Nurbaeti[2], Amalia Shadrina[2,3], Pritta Ameilia Iffanolida[2], Kresna Mutia[2], Ririn Rahmala Febri[2], Retno Asti Werdhani[4], Dwi Anita Suryandari[5], Wisnu Jatmiko[6], Arief Boediono[7,8], Hartanto Bayuaji[9], R. Muharam Natadisastra[1,3], Budi Wiweko[1,2,3]

1 Department of Obstetrics and Gynecology, Faculty of Medicine Universitas Indonesia, Dr.Cipto Mangunkusumo Hospital, Jakarta, Indonesia, 2 Human Reproduction, Infertility, and Family Planning Cluster, Indonesia Reproductive Medicine Research and Training Center, Faculty of Medicine, Universitas Indonesia, Jakarta, Indonesia, 3 Reproductive Immunoendocrinology Division, Department of Obstetrics and Gynecology, Faculty of Medicine Universitas Indonesia, Dr. Cipto Mangunkusumo General Hospital, Jakarta, Indonesia, 4 Department of Community Medicine, Faculty of Medicine, Universitas Indonesia, Jakarta, Indonesia, 5 Department of Biology, Faculty of Medicine, Universitas Indonesia, Jakarta, Indonesia, 6 Faculty of Computer Science Universitas Indonesia, Jakarta, Indonesia, 7 Faculty of Veterinary Medicine Bogor Agricultural University, Bogor, Indonesia, 8 Indonesian Reproductive Science Institute (IRSI), Research and Training Centre, Jakarta, Indonesia, 9 Departement of Obstetrics and Gynecology, Faculty of Medicine Universitas Padjajaran, Bandung, Indonesia

* kemal.achmad@gmail.com

## Abstract

### Background

In Vitro Fertilization (IVF) is one of the most effective infertility treatments. Preimplantation genetic testing for aneuploidy (PGT-A) is invasive, high-cost, and has the potential risk of embryo damage. Our study explores noninvasive approaches to assess embryo quality through morphology and morphokinetics and analyze miRNA, cell-free DNA (cfDNA) for embryo chromosomal status.

### Methods

This cross-sectional study followed the STROBE guideline. We analysed the relationship between embryo morphology, morphokinetics, cfDNA, and microRNAs with the aneuploidy status of day-5 embryos at the blastocyst stage. Non-probability consecutive sampling yielded 124 embryos between 28th December 2021 and 31st December 2022 in our center. Model performance was evaluated using the area under the curve (AUC) from receiver operating characteristic (ROC) analysis.

**Data availability statement:** The data supporting the findings of this study are accessible at https://doi.org/10.5281/zenodo.18089014 [28].

**Funding:** The author(s) received no specific funding for this work.

**Competing interests:** The authors have declared that no competing interests exist.

## Results

Of 124 embryos, 50.8% had aneuploid chromosomes. Analysis of ccfDNA demonstrated a significantly higher rate of aneuploidy from aneuploid embryos compared to the euploid group (p < 0.001). Morphokinetic parameters assessment showed longer pronuclear fading time (21.9 vs. 19.9 hours; p = 0.041) and shorter third cell cycle synchrony (13.4 vs. 17.5 hours; p = 0.036). The expression of miRNA-191 and miRNA-372 was higher in aneuploid compared to euploid embryos (8.1 fold, p < 0.001, and 6.5 fold, p = 0.008, respectively). ROC analysis revealed predictive values for miRNA-191 (AUC 0.745, sensitivity 73%, specificity 70.5%) and miRNA-372 (AUC 0.638, sensitivity 68.3%, specificity 55.7%). MiRNA-191, expansion category, and noninvasive PGT-A (niPGT-A) were identified as independent predictors of embryo ploidy status (p = 0.005).

## Conclusion

Embryo chromosomal status can be evaluated by integrating morphology, morphokinetics, miRNA-191, miRNA-372, and cell-free DNA.

---

## Introduction

In Vitro Fertilization (IVF) has become one of the most effective infertility treatments. It has been widely used over the past decades, accounting for 30–40% of cases in Indonesia between 2019 and 2020, with endometrial receptivity and embryo quality influencing successful implantation and improving pregnancy rates.[1]

Embryo quality can be further evaluated through invasive preimplantation genetic testing for aneuploidy (PGT-A) as a consideration before embryo transfer.[2] Biopsy of blastomeres at the 8-cell stage or of trophectoderm is performed to analyse DNA from oocysts or embryos for determining the aneuploidy status of the embryo. Some studies have not clarified whether PGT-A causes damage to embryos, nor have they addressed the possibility of mosaicism or undetected segmental aneuploidy.[3] The high cost, invasiveness, and potential risk of embryo damage associated with preimplantation genetic testing for aneuploidy (PGT-A) have increased interest in developing simpler, more cost-effective, and noninvasive techniques for predicting embryo aneuploidy. Alternative approaches to assess embryo viability include time-lapse analysis of the embryo, visual assessment of embryo morphology in remaining embryos at day 3 or day 5, and biochemical analysis of cell-free DNA (cfDNA) and exosomal miRNA (particularly miRNA-372 and miRNA-191) as potential biomarkers of embryo viability at the chromosomal level before implantation. A retrospective analysis of 306 embryos showed that euploidy status was closely associated with morphological and morphokinetic features. One related factor to euploidy status is the inner cell mass and trophectoderm of blastocyst.[4]

The embryo visual assessment is a common method for selecting embryos by examining their morphology at a specific time. Certain variables are evaluated at different stages of development to assess embryo quality. Time-lapse microscopy

techniques are used to analyse embryo morphology at regular intervals. Morphokinetic assessment of embryos begins during the early stages, starting with the progressive stages (2-cell, 3-cell, 4-cell, 5-cell, 6-cell, and 9-cell stage) (t1-t9), and later at the time of partial compaction.[5] Traditional morphology assessments are insufficient for reliably detecting chromosomal abnormalities, morphokinetic analysis is often limited by interobserver and interlaboratory variability. Visual assessment of embryos provides more accurate and standardized evaluations.[6–8] Additionally, analysis of microRNA (miRNA) and cell-free DNA (cfDNA) in embryo culture offers a promising noninvasive alternative for determining the embryo's chromosomal status.

This study aims to explore embryo quality assessment based on morphology and morphokinetics and analyze miRNA and cfDNA in the culture medium as predictive tools for embryo chromosomal status, thereby reducing the need for invasive PGT-A procedures.

## Materials & methods

### Study design

This cross-sectional study followed the STROBE guideline. We analysed the relationship between embryo morphology, morphokinetics, cell-free DNA (cfDNA), and microRNAs with the aneuploidy status of day-5 embryos at the blastocyst stage. Research samples consisted of embryo cultures from patients who underwent in vitro fertilization (IVF) and had next-generation sequencing (NGS) done at our center. Samples were collected between December 2021 and December 2022. Subsequent examinations were performed at the PGD Cluster of Human Reproduction, Infertility, and Family Planning, IMERI Faculty of Medicine, Universitas Indonesia (FMUI) Laboratory, as well as at the Diagnostic and Research Centre Biomolecular Laboratory, to analyse chromosomal status using NGS. Non-probability consecutive sampling yielded 124 embryos for analysis.

### Samples and participants selection

Inclusion criteria were IVF patients who underwent NGS, those with at least three embryos graded as A or B, and those who provided written informed consent. Exclusion criteria were embryos that failed to reach the blastocyst stage by day 5 of culture, embryos without available biopsy samples for chromosomal analysis, embryos with undesired mosaicism, and embryos not cultured separately.

For culture media retrieval, metaphase-II oocytes were collected and subjected to intracytoplasmic sperm injection (ICSI). At 18–20 hours post-ICSI, fertilization was confirmed by the presence of oocytes. Oocytes with two pronuclei were cultured up to day 3 and transferred to a phenol red-free blastocyst medium.

### Embryo culture and medium collection

After intracytoplasmic sperm injection (ICSI), fertilized oocytes were individually cultured in separate 20 µl micro drops (Origio) up to the blastocyst stage (day 5–6), and incubated at 37℃ in a humidified atmosphere of 6% $CO_2$ and 5% $O_2$. On day 3, cleavage-stage embryos were transferred to a fresh individual 20 µl drop of medium's name (Origio). Spent embryo media cultures were collected after embryo biopsy on Day 5. Embryo development was monitored daily using morphokinetic time-lapse imaging to evaluate morphological quality. A trophectoderm biopsy was performed, and the remaining culture medium was utilised for miRNA and cfDNA isolation. The media were separated into batch-certified, sterile, PCR (free from DNA, DNase, RNase and PCR inhibitors) tubes (Eppendorf, Hamburg, Germany) and immediately frozen into and stored at -80℃.

### Trophectoderm biopsy

Blastocyst-stage embryos on day 5 were reported to patients, and biopsies for further chromosomal analysis were performed with patient consent. Each embryo was placed in a separate medium, and 4–10 trophectoderm cells were

 

extracted using a laser to create a 6–9 μm opening in the zona pellucida under microscopic guidance. The collected cells were washed in phosphate-buffered saline (PBS) with 1% polyvinylpyrrolidone (PVP), placed into PCR tubes containing 2.5 μL of medium, and stored at –20°C for up to one week before chromosomal analysis via NGS.

Whole genome amplification (WGA) was performed on the TE biopsies using the 24Sureplex DNA Amplification System (Oxford Gene Technology, UK). The amplified products were quantified and normalized for library preparation. Subsequent sequencing and analysis were conducted following the standard NGS workflow. Chromosomal results from the trophectoderm biopsy were classified as euploid or aneuploid based on the software's analysis parameters, which flagged whole-chromosome gains or losses exceeding 80% mosaicism as abnormal. The concordance rate between this invasive PGT-A result and the noninvasive cfDNA analysis was subsequently calculated.

## Cell-free DNA chromosomal analysis

Twenty microliters of media were recovered from each Day 5 embryo culture and transferred to DNA-free/DNase-free tubes under sterile conditions. Basal medium culture was collected as controls. Samples and controls were stored at −80 °C. Due to the low quantity of expected DNA present in the spent culture media, we used the 2.5 uL amount of SBM for WGA, with two consecutive amplification steps performed to increase the sensitivity of the technique, with the Sureplex DNA amplification system (Illumina, USA) was used to double amplify each sample individually.

Next Generation Sequencing (NGS), trophectoderm biopsied, positive control from male human genomic DNA (Promega™ G1471, Wisconsin, USA), negative controls and SBM were first lysed and genomic DNA was fragmented and amplified using 24Sureplex DNA Amplification System (Illumina, Inc., San Diego, CA, USA, PR-40-415101-00), according to the manufacturer's protocol. All the samples (negative and positive controls) were prepared for sequencing using Veriseq library preparation kit as manufacturer protocols (Illumina, Inc., San Diego, CA, USA, RH-101–1001). The samples were then sequenced using a Miseq NGS (Illumina) for 8 hours. Finally, the data from NGS were analyzed using Bluefuse software 2.0 to aneuploidy calling.

## Analysis of MicroRNA expression from spent culture medium

MicroRNA was isolated from spent embryo culture media using the mirVana™ PARIS™ RNA and Native Protein Purification Kit (AM1556, Thermo Fisher Scientific, USA) in accordance with the manufacturer's protocol. Briefly, 20 μL of spent culture medium was subjected to lysis, acid phenol–chloroform extraction, and ethanol-based purification using filter cartridges. Purified miRNA was eluted using preheated elution solution and stored at −80°C until further analysis.

Complementary DNA (cDNA) synthesis was performed using the TaqMan™ MicroRNA Reverse Transcription Kit (4366596, Thermo Fisher Scientific, USA). Reverse transcription reactions were carried out under the following thermal cycling conditions: 16°C for 30 minutes, 42°C for 30 minutes, and 85°C for 5 minutes.

Quantitative real-time PCR (qRT-PCR) was conducted using TaqMan™ Fast Advanced Master Mix and TaqMan™ Advanced miRNA assays to assess the expression levels of miR-191 and miR-372. U6 small nuclear RNA was used as the endogenous control for normalization. Amplification was performed with an initial polymerase activation at 95°C for 20 seconds, followed by 40 cycles of denaturation at 95°C for 3 seconds. All reactions were performed in duplicate. Relative miRNA expression levels were calculated using the comparative threshold cycle (ΔΔCt) method, with results expressed as relative expression values.

## Morphokinetic analysis

Embryo morphokinetic development was assessed using a time-lapse monitoring system throughout the preimplantation period. Continuous imaging was performed to record embryo development without removing embryos from the incubator. Morphokinetic annotations were conducted retrospectively by trained embryologists according to standardized criteria.

The following developmental time points were recorded (in hours): time to pronuclear fading (tPNf), time to reach the 2-cell (t2), 3-cell (t3), 4-cell (t4), 5-cell (t5), 6-cell (t6), 7-cell (t7), and 8-cell (t8) stages, time to start compaction (tSC), time to morula formation (tM), time to start blastocyst formation (tSB), and time to blastocyst formation (tB), expansion grade.

Cell cycle parameters were calculated as follows: the duration of the second cell cycle (cc2), defined as $t3-t2$; the duration of the third cell cycle (cc3), defined as $t5-t3$; the synchrony of the second cell cycle (s2), defined as $t4-t3$; and the synchrony of the third cell cycle (s3), defined as $t8-t5$. All morphokinetic parameters were expressed in hours and analyzed as continuous variables. Embryos with incomplete or ambiguous annotations were excluded from morphokinetic analysis.

### Ethical consideration

This study was approved by the Ethics Committee of the Faculty of Medicine, Universitas Indonesia (KET-1274/UN2. F1/ETIK/PPM.00.02/2021). All procedures adhered to ethical standards, with strict protection of patient confidentiality throughout the research process.

### Statistical analysis

Data analysis included chi-square tests for categorical variables and T-tests or Mann–Whitney tests for numerical variables, depending on distribution normality. The dataset consists of results from artificial intelligence-based embryo morphology assessments, morphokinetic measurements (time to reach the blastocyst stage), miRNA expression, cell-free DNA in the culture medium, and PGT-A biopsy results. The following datasets were tabulated and processed using SPSS 22. Categorical variables for embryo morphology on days 3 and 5 and cell-free DNA (cfDNA) in the sample culture medium are presented as descriptive statistics. Numerical variables, specifically miRNA 372 and 191 expression, were described either in the form of mean ± standard deviation for normal distribution or in the median (interquartile range) for abnormal distribution.

Morphology and cfDNA data were analysed using chi-square tests in relation to PGT-A results. For miRNA expression and morphokinetics, unpaired t-tests were used for normally distributed data, while Mann–Whitney tests were used for non-normal distributions. Variables with $p < 0.250$ in bivariate analysis were included in multivariate logistic regression to identify factors most strongly associated with PGT-A outcomes. Statistical significance was set at $p < 0.05$. Regression coefficients were used to assign weights to predictors, and categorical variables were assessed via B/SE calculations. Scores were derived by dividing each B/SE value by that of the smallest-weight predictor.

Model performance was evaluated using the area under the curve (AUC) from receiver operating characteristic (ROC) analysis, with AUC closer to 1 indicating stronger predictive power. The Hosmer–Lemeshow test was used to assess goodness-of-fit. Sensitivity and specificity values were calculated for non-crossover models to determine diagnostic accuracy.

### Results

Of the 148 medium samples collected, 24 embryo cultures at the blastocyst stage (day 5) presenting with chromosomal mosaicism were excluded based on exclusion and inclusion criteria, resulting in 124 eligible samples from 57 couples, of which 50.8% had aneuploid chromosomes. Regarding embryo morphology, 48.4% of the embryo samples showed day-3 morphological characteristics classified as category A. Assessment of the inner cell mass (ICM) morphology revealed that 54% fell into category A, 33.9% into category B, and 12.1% into category C. Meanwhile, trophectoderm (TE) morphology assessment showed 37.9% in category A, 40.3% in category B, and 21.8% in category C.

The results of bivariate analysis demonstrated no significant association between ICM morphology (OR 1.48; 95%, CI: 0.73–3.02, $p = 0.18$) and TE morphology (OR 1.12; 95% CI: 0.54–2.33, $p = 0.444$) with the chromosomal status of embryos. Odds ratio indicated the likeliness of an embryo with a given ICM or TE grade to be euploid. However, the

confidence interval included 1.0 and the p-values were not significant. These findings showed that morphological parameters failed to predict ploidy status. On the other hand, there was a significant discrepancy between the ploidy status determined from cell-free DNA and the chromosomal status of embryos obtained through PGT-A. Specifically, the euploid cfDNA group exhibited a statistically lower rate compared with the aneuploid cfDNA group (82.2% vs. 98%; OR = 230.76, 95% CI: 29.17–1825.6; p<0.001).

Morphokinetic parameters were assessed using a time-lapse microscope, to record nuclear and cytoplasmic changes in the culture medium from early cleavage (2–9 cells) through partial compaction of embryo, thereby determining cleavage timing and progression to the blastocyst stage (tB). The average time for the start of blastulation (tSB) and time to reach blastocyst stage (tB) were 100.49±10.19 hours (91.2–111.57) and 107.19 hours (91.2–140.20), respectively.

Expansion grade in our bivariate analysis showed that among blastocysts with expansion degree 1–2, 23 of 61 embryos (34.3%) were PGT-A euploid and 44 of 63 embryos (65.7%) were aneuploid. In contrast, for expansion degree 3–6, 38 embryos (66.7%) were euploid and 19 embryos (33.3%) were aneuploid. The blastocysts with higher expansion degree (3–6) had 3.8 fold higher odds of euploidy compared to lower expansion degree (1–2) (66.7% vs. 34.3%; OR 3.82, 95% CI: 1.81–8.07; *p*<0.001)

Morphokinetic parameters, microRNA expression, and chromosomal status from PGT-A revealed significant differences in the time to pronuclear fading, with means of 21.91 hours for aneuploid embryos and 19.88 hours for euploid embryos (95% CI: –3.88– −2.72; p=0.041). Additionally, synchrony of the third cell cycle was significantly associated with PGT-A results, with a mean of 17.46 hours in euploid embryos and 13.43 hours in aneuploid embryos (95% CI: −4.45– −2.90; p=0.036). However, no statistically significant associations were observed between abnormal bivariate morphokinetic parameters based on direct cleavage, multinucleation, and reverse cleavage with PGT-A chromosomal status (p>0.05).

Our study then examined the multivariate analysis in more detail. An ROC curve was then performed to assess sensitivity, specificity, and cut-off values for miRNA-191 and miRNA-372 expression levels indicating higher miRNA-191 expression was associated with a greater risk of aneuploid chromosomes. The area under the curve (AUC), cut-off value, sensitivity, and specificity of miRNA-372 expression were 0.638, 4.07, 68.3%, and 55.7% respectively. The AUC, cut-off value, sensitivity, and specificity of miRNA-191 expression were 0.745, 3.745, 73% and 70.5%, respectively. Bivariate analysis reported that the miRNA-191 expression in the PGT-A aneuploid group with the cut-off value ≥3.745 was significantly higher than PGT-A euploid group (OR 6.446; 95% CI: 2.956–14.136, p<0.001; n=71,9%).

The cut-off value was used as the critical threshold to categorize miRNA expression levels. Values equal to or exceeding the cut-off were classified as high expression for miRNA-372 and miRNA-191 in association with euploid embryo status, while values below the cut-off were designated as low expression. Bivariate analysis revealed that the high-expression group of miRNA-191 in aneuploid embryos, as determined by PGT-A, had significantly higher expression than the low-expression group (p <0.001). In our study, miRNA-191 and miRNA-372 expression increased by 3.86-fold and 4.87-fold, respectively. Further analysis based on embryonic chromosomal status revealed an 8.11-fold increase in miRNA-191 expression in the culture medium of aneuploid embryos. Similarly, miRNA-372 expression was elevated by 6.45 and 3.53 fold in aneuploid and euploid embryos.

Median expression levels of miRNA-191 and miRNA-372 were 3.86 and 4.87 fold higher, respectively, than those of control genes. Bivariate analysis indicated that miRNA-191 expression was significantly elevated in aneuploid embryos (8.11 fold, p<0.001) compared to euploid embryos. A comparable trend was observed for miRNA-372, with a median 6.45-fold increase in aneuploid embryos (p=0.008).

Multivariate analysis identified miRNA-191, expansion category, and noninvasive preimplantation genetic testing for aneuploidies (niPGT-A) as statistically significant predictors across seven modeling steps. In the final three steps, miRNA-191, niPGT-A, and the expansion category remained substantial (95% CI: 1.95–48.36, p=0.005). The model demonstrated strong explanatory power (Nagelkerke R²=0.800) and good fit (Hosmer-Lemeshow test, p>0.817). Though

bivariate analysis showed miRNA-372 was elevated in aneuploid embryos, it did not retain independent predictive value in the multivariate model (p > 0.05) and was not included in the final model.

We deduced the final weighted scoring prediction model based on the scores for each of the variables in Table 1, to predict embryo ploidy. The model integrates several noninvasive parameters with each one assigned a points value based on its predictive strength determined by logistic regression. The blastocyst expansion category was divided into six categories according to Gardner's Criteria. The scores for categories 1–2 were 0, and the scores for categories 3–6 were 1. NiPGT-A for the embryo euploid status was scored as 2, and aneuploidy was scored as 0. The relative miRNA-191 expression below our cut-off value (<3.745) was scored as 1. By simply adding the points from blastocyst expansion, niPGT-A results, and miRNA-191 levels, we arrive at a composite score. This offers a practical and quantifiable method to assess chromosomal status, which provides a useful tool to select embryos in IVF.

A closer look at the scoring in Table 2 shows that niPGT-A carries more weight than the others. Where most morphological and molecular markers are scored on a simple 0 or 1 basis, niPGT-A is assigned a value of 2 for a euploid result. This stemmed from the statistical analysis, which identified niPGT-A as a substantially stronger predictor. Its regression coefficient was so pronounced that, when standardized, it warranted a double score. This effectively positions the niPGT-A result as a more decisive factor in the final calculation, highlighting its critical role in the model's assessment of ploidy.

An analysis was performed to determine the cut-off point, as shown in **Fig 1**. The prediction model using the categorical variables of expansion stage, niPGT-A, and miRNA-191 secretion produced a cut off value of >2.5, with a sensitivity of 91.8% and a specificity of 88.9% for predicting euploid embryos.

## Discussion

The primary objective of this study was to analyse further comparable parameters obtained from embryo culture medium and invasive preimplantation genetic testing for aneuploidy (PGT-A), to provide data to support decision-making in predicting embryonic chromosomal status using a noninvasive approach as an alternative to invasive PGT-A. The evaluated parameters included embryo morphology and morphokinetics, the expression levels of secreted miRNA-372 and miRNA-191, and cell-free DNA in the culture medium. Our findings demonstrated a significant association between embryo quality grades and ploidy status. Embryos of higher quality showed greater predictive value for implantation potential and live birth rates compared to blastomere counts, particularly in cleavage-stage embryos, and outperformed the corresponding outcomes in conventional IVF subjects.[9]

Table 1. Calculation of variable prediction model.

| Variable | B | SE | Odds Ratio | 95% CI for Odds Ratio | p value | B/SE | Model Score |
|---|---|---|---|---|---|---|---|
| **Expansion Category (3–6)** | 0.113 | 0.776 | 8.271 | 1.808 - 37.837 | 0.006 | 2.723 | **1** |
| **NiPGT-A (euploid)** | 5.695 | 1.165 | 286.980 | 29.268 - 2813.868 | <0.001 | 4.857 | **2** |
| **miRNA-191** | 2.140 | 0.748 | 8.498 | 1.963 - 36.794 | 0.004 | 2.86 | **1** |

Table 2. Variable score on prediction model 1.

| Variable | Category / Cut-off | Score |
|---|---|---|
| **Expansion Category** | 1-2 | 0 |
| | 3-6 | 1 |
| **niPGT-A** | Aneuploid | 0 |
| | Euploid | 2 |
| **miRNA 191 relative expression** | < 3.745 | 1 |
| | ≥ 3.745 | 0 |

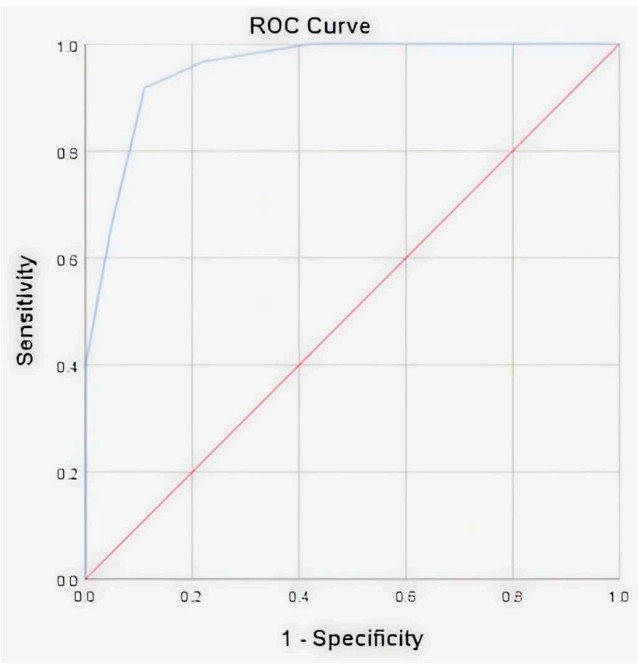

**Fig 1. ROC curve of prediction model 1 incorporated expansion category, miRNA-191 and niPGT.-A.**

Of the 124 data samples analyzed, day-3 embryo morphology was categorized as grade A in 48.4% and grade B in 49.2%. Good inner cell mass (ICM) morphology (grade A) was observed in 54%, while trophectoderm morphology was grade A in 37.9% and grade B in 40.3% of cases. These results align with a retrospective study of embryo transfers over a five-year period, which reported the majority of embryos falling into the AA group (35.8%) and BB group (35.3%) based on trophectoderm morphology, with lower representation in the CA category.[10] In our cohort, only 2.4% were classified as category C. Similarly, a retrospective cohort study involving 652 subjects and 3,238 biopsied blastocysts following PGT for monogenic disorders (PGT-M) demonstrated a strong correlation between embryo morphology and chromosomal ploidy status. C-grade ICM morphology, as well as B- or C-grade trophectoderm morphology, was less significantly associated with euploidy, whereas blastocyst morphology overall showed a strong correlation with euploidy status (up to 73.2%).[11]

Conversely, another cohort study reported no significant association between euploidy status and blastocyst developmental stage. In that study, euploid embryo morphology was not closely linked to live birth rates; however, the odds of live birth were significantly higher for blastocysts with A-grade trophectoderm compared to those with C-grade trophectoderm (62.71% vs. 45.40%; OR 2.212, 95% CI: 1.164–4.201; p = 0.015).[12] Additionally, live birth rates were reported to be higher for day-5 blastocysts compared to day-6 euploid blastocysts (57.75% vs. 41.67%; OR 2.247, 95% CI: 1.460–3.460; p < 0.001).

## Cell-free DNA (cfDNA) and chromosome status

Previous studies have demonstrated that cell-free DNA (cfDNA) obtained from spent embryo culture medium can serve as a potential screening tool for determining chromosomal status, including the identification of poor-quality blastocysts. Using our standardized protocol (detailed in Methods), we isolated and analyzed cfDNA from spent culture media. This methodological alignment between invasive and noninvasive approaches allowed for direct comparison, yielding an overall concordance rate of 82.2% between niPGT-A and invasive PGT-A for euploid embryos. Invasive embryo biopsy

remains widely used for chromosomal assessment; however, its inherent drawbacks highlight the need for alternative approaches. A noninvasive option is cfDNA analysis from a spent culture medium.

In one study, the concordance between trophectoderm biopsy and spent culture medium was relatively low, with only 20.8% agreement for monosomy cases. The detection rate of cfDNA from spent culture medium was 56.3% on day 3, increasing to 65% by day 5 of embryo development. For euploid embryos, cfDNA obtained from spent culture medium demonstrated 82.2% concordance with chromosomal status as determined by PGT-A.[13] This aligns closely with our observed concordance, though it is lower than the 89.9% reported by Huang et al. which may be attributed to differences in culture medium volume, collection timing, or bioinformatic thresholds.[14] This limitation of trophectoderm biopsy is reflected in reported PGT-A accuracy rates ranging from 62.1% to 86.2%, showing its inability to represent the entire genomic profile of the embryo.

Another study reported niPGT-A testing with a sensitivity of 100%, and a specificity of 87.5% (PPV 88.9%, NPV 100%, FPR 12.5% and FNR 0%). Meanwhile, PGT-A testing showed a sensitivity of 87.5%, and a specificity of 77.8% (PPV 87.5%, NPV 75%, FPR 14.3% and FNR 22.2%). niPGT-A had better ploidy diagnostic value compared to PGT-A. A positive correlation of 89.9% between the morphology of the inner cell mass and the trophectoderm of the blastocyst was also reported.[15] While our niPGT-A performance was robust, the sensitivity and specificity of our final integrated model (91.8% and 88.9%, respectively) surpass many standalone niPGT-A reports, underscoring the additive value of combining molecular and morphological parameters.

## Expression of miRNA-372 and miRNA-191 and chromosome status

In our study, high miRNA-191 expression was more prominently observed in embryos with aneuploidy of PGT-A-derived chromosome status. In a cohort study of 28 embryos, miRNA-191 was 4.7-fold more highly expressed in culture media from aneuploid embryos (p = 0.031). MiRNA-191 and miRNA-372 were 5.1-fold more highly concentrated in media from failed IVF or non-intracytoplasmic sperm injection cycles.[16] Our findings revealed an 8.11-fold increase in miRNA-191 expression in aneuploid compared to euploid embryos, which is notably higher than the 4.7-fold increase reported in a prior cohort study. This stronger association may be due to our larger sample size (n = 124) or optimized miRNA isolation from the dedicated media aliquot. Out of the findings, miRNA-191 may serve as a biomarker of embryo aneuploidy and pregnancy failure. Meanwhile, while bivariate analysis showed miRNA-372 was elevated in aneuploid embryos (6.45-fold), high miRNA-372 expression was not correlated with the ploidy status of the embryo in our final multivariate model, a finding that contrasts with some earlier reports and may reflect differences in patient population or the superior specificity of miRNA-191. miRNA-191 and miRNA-372 regulate the mitogen-activated protein kinase kinase kinase 1 (MAP3K1) and cyclin-dependent kinase 6 (CDK6), genes critical in cell cycle, signaling, and apoptotic pathways.[17,18]

CDK6 was the bona fide target of miRNA-191, disrupting the cell cycle. miRNA-191 represses CDK6, which affects maturation proliferating factor, resulting in chromosomal instability and aneuploidy. Following DNA destruction, upregulated miRNA-191 played a role in aneuploid chromosomes. [17,18] This mechanistic pathway supportedcel the strong association we observed between elevated miRNA-191 and chromosomal abnormalities.

Several miRNA expression levels have been shown to play crucial roles in the development of mammalian embryos and the maintenance of stem cell pluripotency. Several differentially expressed miRNAs were found based on chromosomal arrangement.[19,20] Previous studies also reported that miRNA-372 was detected in IVF failures, with the largest factor being implantation failure. Due to embryonic factors, miRNA-372 was the most expressed in euploid embryos. Although miRNA-372 was considered to play a role in chromosome status, current research on this association remains limited. Therefore, it explained why its specificity on chromosome status was lower than that of miRNA-191. It was previously reported that miRNA-191 was more concentrated in media from aneuploid embryos. miRNA-191, miRNA-372 and miRNA-645 were also found to be more concentrated in media from failed IVF cycles. In a study comparing embryo cleavage stage and implantation outcomes with miRNAs in culture media and sperm, it was found that miR-320a, as well

as miR-19b-3p, miR-15a-5p, miR-21-5p, and miR-20a-5p were differentially expressed. A study comparing miRNA expression in human follicular fluid showed that miRNA-320a levels were significantly different in high-quality embryos compared to non-quality embryos at day 3.[21]

We focused on miRNA-191 and miRNA-372 due to their previously reported associations with embryo viability and chromosomal status. The activities of hsa-miR-191-5p on endometrial markers in the implantation window and hsa-miR-24–1-5p on cell proliferation and migration are represented in the model.[19] mir-191 is upregulated in the culture media of implanted human embryos, resulting in a 5.2 fold greater expression of mir-191 in the culture media from cycles achieving pregnancy compared with non-pregnant patients.[15]

The expression of hsa-miR-191-5p also modulates various proteins, two of which are insulin-type growth factors (IGF2 BP-1 and IGF2R) associated with decidualisation of endometrial tissue. According to the study, miRNAs are not only potential biomarkers of implantation viability, but can also be secreted into the extracellular environment to induce cell activation in favour of embryo implantation and development.[15,22] Thus, hsa-miR-191-5p needs to be considered.

Our study supports previous research indicating that assessing embryo quality in IVF can increase the implantation success rate. Moreover, miRNAs are considered to provide a general overview of embryo–maternal communication, thus potentially serving as noninvasive markers of embryo quality. This approach could improve assessment accuracy and reduce the risk of mechanical injury to the embryo.

### Role of miRNA-191, expansion grade, morphokinetics and Cell-free DNA parameters in prediction model of embryo chromosome status

A study showed time to pronuclear fading (tPNf) was significantly slower in abnormal embryos (p < 0.05).[23] In this study, a slower tPNf was obtained in the aneuploid embryo group. Amir et al. in their study said embryos carrying non-viable translocations, especially those of maternal origin, showed a significant delay in pronuclei loss. In another study, it was found that a faster time to pronuclei loss can indicate the potential for good quality embryos and result in a higher number of live births. The use of time-lapse imaging in embryos can help ensure increased accuracy of pronuclei assessment. It also allows visualisation of transient pronuclei that may form and disappear slightly earlier or later than normal, thereby helping to ensure selection of euploid embryos for transfer.[24]The duration of time to initiate morula compaction or time to compact (tcom) is also known as the initial process of fusion of two cells. The mechanism linking normal embryos and timing of morula compaction remains unexplained. Prior to morula compaction, cleavage-stage embryos consist of relatively equal and independent blastomeres. Each blastomere then flattens its membrane against the others and rapidly increases cell-to-cell contact due to increased expression of E-cadherin as an adhesion protein. This occurrence allows the morula compaction process to serve as an early indicator of embryonic gene expression. Several studies have reported a relationship between compaction and embryo quality, finding that the timing pattern of morula compaction influenced embryo implantation rates and compaction that began before the eight-cell stage was usually associated with poor embryo development. During the compaction process, a self-correction mechanism can also occur by excluding abnormal blastomeres.[25,26]

Our study reported the degree of expansion as a morphological parameter associated with the embryo's chromosome status. Logistic regression analysis by Kim et al. found that the most influential morphological factor was the degree of expansion, compared with the degree of inner cell mass or trophectoderm (OR = 1.261, p = 0.045).[27] Our model assigned an odds ratio of 8.271 to expansion categories 3–6, a substantially stronger effect size that may reflect our strict grading criteria and cohort characteristics. The study by Huang et al. explained the possibility that the aneuploid state can functionally interfere with the ability of the differentiated trophectoderm epithelium to expand productively, so in conclusion, the higher the degree of expansion, the higher the percentage of the possibility of euploid blastocysts.[11,14]

In summary, our integrated prediction model, which combines miRNA-191, expansion category, and niPGT-A, demonstrates strong diagnostic performance. The model's AUC of 0.800 (Nagelkerke $R^2$) and high sensitivity/specificity compare

favorably to models relying on single parameters, as reported in recent literature. This quantitative improvement supports the clinical utility of a multi-modal, noninvasive assessment strategy for embryo selection.

## Strengths and limitations

To our knowledge, our research study provides a comprehensive assessment of morphological and morphokinetic parameters, miRNA 372 and 191 secretions, and cell-free DNA, thus contributing to the prediction of the chromosomal status of embryos.

The study included patients undergoing IVF, with embryo culture media collected on day 5, subsequently analysed for chromosomal composition using Next Generation Sequencing (NGS). To reduce confounding variables, the inclusion criteria required a minimum number of high-quality embryos (grade A and B) on day 3, thereby establishing baseline embryo quality. Exclusion criteria eliminated embryos that failed to advance to the blastocyst stage or exhibited mosaicism, thereby reducing variability resulting from atypical developmental stages that could distort chromosomal status evaluation.

The study recognizes that its predictive models for embryo chromosomal status did not explicitly account for the substantial confounding variables of maternal/paternal age and semen quality. Although the models included embryo-specific markers (morphology, morphokinetics, miRNA and cfDNA) and employed statistical techniques to mitigate certain confounders, this is acknowledged as a significant limitation. Further research is also warranted to externally validate noninvasive chromosomal status prediction models using a larger number of samples.

## Conclusion

Our research study is the first to conduct a comprehensive assessment of embryo quality, including morphology, morphokinetics, miRNA secretion levels, and cell-free DNA. Moreover, our study provides prediction models, achieving sensitivity and specificity through logistic regression-based model.

## Acknowledgments

The authors thank all the participants and supervisors who contributed to the study.

## Author contributions

**Conceptualization:** Achmad Kemal Harzif.

**Data curation:** Achmad Kemal Harzif, Intan Indah Permatasari, Patricia Alika Kurniawan, Thalia Amila Elsiyana, Pritta Ameilia Iffanolida, Kresna Mutia, Ririn Rahmala Febri.

**Formal analysis:** Achmad Kemal Harzif, Intan Indah Permatasari, Patricia Alika Kurniawan, Thalia Amila Elsiyana, Pritta Ameilia Iffanolida, Kresna Mutia, Ririn Rahmala Febri.

**Funding acquisition:** Achmad Kemal Harzif.

**Investigation:** Achmad Kemal Harzif.

**Methodology:** Achmad Kemal Harzif, Intan Indah Permatasari, Patricia Alika Kurniawan, Thalia Amila Elsiyana, Fistyanisa Elya Charilda, Azizah Fitriayu Andyra, Heidi Dewi Mutia, Nafi'atul Ummah, Putri Nurbaeti, Amalia Shadrina.

**Project administration:** Achmad Kemal Harzif.

**Resources:** Intan Indah Permatasari, Patricia Alika Kurniawan, Thalia Amila Elsiyana, Fistyanisa Elya Charilda, Azizah Fitriayu Andyra, Heidi Dewi Mutia, Nafi'atul Ummah, Putri Nurbaeti, Amalia Shadrina.

**Software:** Intan Indah Permatasari, Patricia Alika Kurniawan, Thalia Amila Elsiyana, Fistyanisa Elya Charilda, Azizah Fitriayu Andyra, Heidi Dewi Mutia, Nafi'atul Ummah, Putri Nurbaeti, Amalia Shadrina, Retno Asti Werdhani, Dwi Anita Suryandari, Wisnu Jatmiko, Arief Boediono, Hartanto Bayuaji, R. Muharam Natadisastra, Budi Wiweko.

**Supervision:** Achmad Kemal Harzif.

**Validation:** Achmad Kemal Harzif, Fistyanisa Elya Charilda, Azizah Fitriayu Andyra, Heidi Dewi Mutia, Nafi'atul Ummah, Putri Nurbaeti, Amalia Shadrina.

**Visualization:** Intan Indah Permatasari, Patricia Alika Kurniawan, Thalia Amila Elsiyana, Fistyanisa Elya Charilda, Azizah Fitriayu Andyra, Heidi Dewi Mutia, Nafi'atul Ummah, Putri Nurbaeti, Amalia Shadrina, Retno Asti Werdhani, Dwi Anita Suryandari, Wisnu Jatmiko, Arief Boediono, Hartanto Bayuaji, R. Muharam Natadisastra, Budi Wiweko.

**Writing – original draft:** Achmad Kemal Harzif, Intan Indah Permatasari, Patricia Alika Kurniawan, Thalia Amila Elsiyana.

**Writing – review & editing:** Achmad Kemal Harzif, Intan Indah Permatasari, Patricia Alika Kurniawan, Thalia Amila Elsiyana, Fistyanisa Elya Charilda, Azizah Fitriayu Andyra, Heidi Dewi Mutia, Nafi'atul Ummah, Putri Nurbaeti, Amalia Shadrina, Pritta Ameilia Iffanolida, Kresna Mutia, Ririn Rahmala Febri, Retno Asti Werdhani, Dwi Anita Suryandari, Wisnu Jatmiko, Arief Boediono, Hartanto Bayuaji, R. Muharam Natadisastra, Budi Wiweko.

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
