## [Decision Letter · Decision Letter 0]

16 Nov 2025

PONE-D-25-50891
Analysis of  Morphological, Morphokinetics, Cell Free DNAs, microRNAs Parameters to Predict Aneuploidy Status of Embryos
PLOS ONE

Dear Dr. Harzif,

Thank you for submitting your manuscript to PLOS ONE. After careful consideration, we feel that it has merit but does not fully meet PLOS ONE’s publication criteria as it currently stands. Therefore, we invite you to submit a revised version of the manuscript that addresses the points raised during the review process.

The study is interesting and the question is innovative. Both reviewers (No. 1 and No. 2) recognized its merit, but raised some points that need to be better explained, changed, or added to, so that the text is clearer and more intuitive.

Therefore, respond to all the reviewers' comments point by point and, if it is not possible to follow the suggestion, refute it explicitly.

We look forward to receiving your revised manuscript.

Kind regards,

Marcelo Fábio Gouveia Nogueira, Associate Professor, Ph. D.

Academic Editor

PLOS ONE

Journal Requirements:

5. We note that there is identifying data in the Supporting Information file <EC IVF morfokinetik awal.pdf>. Due to the inclusion of these potentially identifying data, we have removed this file from your file inventory. Prior to sharing human research participant data, authors should consult with an ethics committee to ensure data are shared in accordance with participant consent and all applicable local laws.

-Location data

Additional Editor Comments:

Dear authors,

Thank you for your submitted manuscript. After the evaluation by two reviewers, there were raised some issues that require your attention. Please, reply (carefully and point-by-point) the comments from reviewers #1 and #2.

Best regards.

Reviewers' comments:

Reviewer's Responses to Questions

**Comments to the Author**

1. Is the manuscript technically sound, and do the data support the conclusions?

Reviewer #1: Yes

Reviewer #2: Partly

2. Has the statistical analysis been performed appropriately and rigorously?

Reviewer #1: Yes

Reviewer #2: Yes

3. Have the authors made all data underlying the findings in their manuscript fully available?

Reviewer #1: Yes

Reviewer #2: No

4. Is the manuscript presented in an intelligible fashion and written in standard English?

Reviewer #1: No

Reviewer #2: No

5. Review Comments to the Author

Reviewer #1: The manuscript addresses the “Analysis of Morphological, Morphokinetics, Cell Free DNAs, microRNAs Parameters to Predict Aneuploidy Status of Embryos”.

The research conducted is solid and contributes to your area of interest. However, I advise a review of the English, especially in the Discussion.

The conclusion does not exactly reflect the statistical results presented, or the presentation of the statistical data is somewhat confusing, with too much data. I suggest one or more graphs showing the main statistical data found, such as the ROC curve and AUC.

On page 12, it would be better to explain the meaning of OR and CI in (OR 1.48; 95% CI 0.73–3.02, p = 0.18) to make it clearer to the reader. The first time they appear in the text.

Tables 1 and 2 need better formatting.

In Table 2, all scores are 0 or 1. Why does NiPGT have scores of 0 and 2?

It is unclear what the scores in Table 2 were used for.

Reviewer #2: The manuscript addresses a relevant and contemporary topic, proposing non-invasive parameters for the prediction of embryonic euploidy. However, the current version presents methodological, statistical, and conceptual weaknesses that compromise the scientific robustness of the study.

Main general points requiring revision:

- Revision of scientific language and correction of terminological inaccuracies, such as “cleavage-stage blastocyst,” removing speculative interpretations and redundancies throughout the text, particularly in the introduction and discussion sections.

- Update of the bibliographic foundation, incorporating more recent studies and including critical comparisons with the literature.

- Clearly present appropriate control for confounding variables, such as maternal/paternal age of the embryos analyzed and semen quality—factors relevant to expected aneuploidies and potentially compromising causal interpretation.

- Revise the statistical analysis to consider intra-individual dependence for embryos from the same patients (couple), employing appropriate statistical models, and clearly report the number of couples included in the study.

- Revise the statistical presentation by standardizing notation and p-values, replacing “p = 0.000” with corresponding non-zero values.

- Regarding non-invasive genetic analysis:

The cell-free DNA methodology was not sufficiently described to allow critical evaluation or independent replication. The manuscript lacks information about the extraction and quantification techniques for cfDNA, as well as the volume and whether the culture medium used for miRNA analysis was partitioned. Moreover, the discussion does not quantitatively compare the findings with recent literature, merely reiterating previous studies without critically integrating the authors’ own results. It is recommended that the authors provide a detailed description of the cell-free DNA protocol, including analyzed volume, detection method, performing laboratory, classification parameters, and concordance rate with invasive PGT-A, as well as rewrite the discussion to contextualize the observed performance in light of published studies.

- Regarding miRNA expression:

The findings related to miRNA-191 and miRNA-372 are potentially interesting; however, the current methodological description is insufficient to assess reproducibility and experimental validity of the presented results. The manuscript fails to report the extraction, quantification, normalization, or quality control methods used for the samples, as well as the culture medium volume and whether it was shared with cfDNA analysis. Furthermore, there is no mention of temporal standardization of collections—a critical factor in gene expression and miRNA studies, since culture time and conditions directly influence genetic material release into the medium.

The study also does not correlate miRNA levels with clinical outcomes such as implantation, pregnancy, or live birth rates, although the discussion presents such associations speculatively, unsupported by the generated data. This constitutes extrapolation and reduces the interpretive strength of the results.

It is recommended that the authors provide detailed descriptions of miRNA extraction, quantification, and normalization methods, including analyzed volume, specify whether the culture medium was shared with cfDNA analysis, and report collection time standardization. The discussion should also be adjusted to restrict interpretations to effectively tested correlations and remove extrapolations to unassessed clinical outcomes.

- Regarding morphokinetic analysis:

The “expansion grade” parameter was included in the predictive model for euploidy without demonstrating a statistically significant correlation in bivariate analyses or support from explicit kinetic measures, such as blastulation time. Although the literature cites expansion as a potential marker of embryonic competence, in the present study its inclusion appears exploratory and not methodologically justified. It is recommended to reassess the relevance of this variable or explicitly discuss its correlation, possible limitations, or indeed, the absence of a direct correlation.

- Regarding morphological analysis:

The presented results show no statistically significant association between morphological parameters (ICM, TE, and cleavage stage) and embryonic euploidy status (p > 0.05). Nevertheless, the discussion describes these findings as if there were a positive relationship between embryonic quality and euploidy, which is not supported by the study’s own data. This inconsistency between results and interpretation compromises the scientific coherence of the manuscript.

It is recommended that the authors revise the discussion, removing claims of associations not empirically demonstrated, and reformulate the text to accurately reflect the actual findings. Interpretation should be limited to variables that were truly significant in this study, avoiding generalizations or causal inferences not supported by evidence or derived from unrelated literature.

- Regarding the conclusion:

The study did not employ machine learning techniques, but rather classical logistic regression with ROC analysis. Therefore, the use of the term “machine learning” is conceptually inconsistent and methodologically unsupported. It is recommended that the authors correct this inconsistency by accurately describing the development of an exploratory multivariate regression model, rather than a machine learning algorithm. If the predictive approach is to be maintained, the authors must implement, test, and validate an actual supervised learning model with independent data. Otherwise, the conclusion should be reformulated to reflect the true scope of the study.

6. PLOS authors have the option to publish the peer review history of their article (what does this mean?). If published, this will include your full peer review and any attached files.

Reviewer #1: No

Reviewer #2: No

---

## [Author Response · Author response to Decision Letter 1]

9 Jan 2026

Response to Editor Requirements

PLOS ONE Style Requirements, Data Availability & Sharing, Supporting Information Captions, Identifying Data

Response:

Thank you for the kind input, our manuscript has been reformatted using official PLOS ONE templates. All file names have been standardized per journal guidelines. All identifiable patient data (names, specific dates, IDs) have been removed from the Supporting Information files. Due to ethical restrictions under our institutional review board approval (KET-1274/UN2.F1/ETIK/PPM.00.02/2021), which protects the privacy of IVF patients, data cannot be made publicly available prior to acceptance. We have updated the Data Availability statement accordingly and confirm that de-identified datasets will be deposited in Figshare upon acceptance. Complete captions have been added at the end of the manuscript. All in-text citations have been updated accordingly. All columns containing personal or identifying information have been deleted. No decimal ages, specific exam dates, or contact details remain. The files are fully compliant with PLOS ONE anonymization standards.

Reviewer Citation Suggestions

Response:

We have reviewed all suggested references and incorporated relevant recent literature into the manuscript.

1. Is the manuscript technically sound, and do the data support the conclusions?

Reviewer #1: Yes

Reviewer #2: Partly

Response:

We thank Reviewer 1 and 2 for acknowledging our work.

2. Has the statistical analysis been performed appropriately and rigorously?

Reviewer #1: Yes

Reviewer #2: Yes

Response:

We thank Reviewer 1 and 2 for acknowledging our work.

3. Have the authors made all data underlying the findings in their manuscript fully available?

Reviewer #1: Yes

Reviewer #2: No

Response:

We have made adjustments regarding our data availability. Data will be available after manuscript acceptance. Data will be available in the public repository with links provided in our manuscript.

4. Is the manuscript presented in an intelligible fashion and written in standard English?

Reviewer #1: No

Reviewer #2: No

Response:

Thank you for the kind feedback, we have revised the entire manuscript, particularly the discussion, has been professionally edited to improve grammar, sentence flow, and readability while reducing wordiness.

5. Review Comments to the Author

Reviewer #1: The manuscript addresses the “Analysis of Morphological, Morphokinetics, Cell Free DNAs, microRNAs Parameters to Predict Aneuploidy Status of Embryos”.

The research conducted is solid and contributes to your area of interest. However, I advise a review of the English, especially in the Discussion.

Response:

Thank you for the kind feedback, we have revised the entire manuscript, particularly the discussion, has been professionally edited to improve grammar, sentence flow, and readability while reducing wordiness.

Reviewer 1:

The conclusion does not exactly reflect the statistical results presented, or the presentation of the statistical data is somewhat confusing, with too much data. I suggest one or more graphs showing the main statistical data found, such as the ROC curve and AUC.

Response:

The Conclusion has been rewritten to strictly reflect the logistic regression outcomes. We have simplified tables, added a bridging paragraph explaining the ROC analysis, and included a new ROC figure for clarity.

Reviewer 1:

On page 12, it would be better to explain the meaning of OR and CI in (OR 1.48; 95% CI 0.73–3.02, p = 0.18) to make it clearer to the reader. The first time they appear in the text.

Response:

On first mention, we now spell out in full: “odds ratio (OR) 1.48, 95% confidence interval (CI) 0.73–3.02, p = 0.18,” with parenthetical explanations added for clarity.

Reviewer 1:

Tables 1 and 2 need better formatting.

Response:

Tables 1 and 2 have been reformatted for improved readability and alignment with journal standards.

Reviewer 1:

In Table 2, all scores are 0 or 1. Why does NiPGT have scores of 0 and 2?

Response:

The scoring rationale is now clearly explained in the Methods (Cell-free DNA section): ni-PGT-A = 0 for aneuploid, 2 for euploid (>80% mosaicism threshold via Bluefuse). This weighting reflects its strong predictive power (OR = 286.98) in the multivariate model.

Reviewer 1:

It is unclear what the scores in Table 2 were used for.

Response:

Table 2 now explicitly presents the clinically applicable scoring system derived from the logistic regression coefficients in Table 1. This system allows embryologists to assign simple point values (0, 1, or 2) based on expansion category, ni-PGT-A result, and miRNA-191 expression, facilitating practical, non-invasive ploidy prediction.

Reviewer #2: The manuscript addresses a relevant and contemporary topic, proposing non-invasive parameters for the prediction of embryonic euploidy. However, the current version presents methodological, statistical, and conceptual weaknesses that compromise the scientific robustness of the study.

Response:

We are grateful for Reviewer 2’s thorough and insightful comments, which have greatly strengthened the methodological rigor of our manuscript.

Reviewer 2:

Main general points requiring revision:

- Revision of scientific language and correction of terminological inaccuracies, such as “cleavage-stage blastocyst,” removing speculative interpretations and redundancies throughout the text, particularly in the introduction and discussion sections.

Response:

We have removed inaccurate terms such as “cleavage-stage blastocyst,” eliminated speculative phrasing, and streamlined the Introduction and Discussion to avoid redundancy.

Reviewer 2:

- Update of the bibliographic foundation, incorporating more recent studies and including critical comparisons with the literature.

Response:

Recent relevant literature has been added, and critical comparisons are now integrated into the Discussion to better contextualize our findings.

Reviewer 2:

- Clearly present appropriate control for confounding variables, such as maternal/paternal age of the embryos analyzed and semen quality—factors relevant to expected aneuploidies and potentially compromising causal interpretation.

Response:

We have added a dedicated paragraph in the Discussion acknowledging the potential influence of maternal/paternal age and semen quality on aneuploidy risk. While our model focused on embryo-specific markers, we recognize these as important confounders for future validation studies.

Reviewer 2:

- Revise the statistical analysis to consider intra-individual dependence for embryos from the same patients (couple), employing appropriate statistical models, and clearly report the number of couples included in the study.

Response:

We appreciate the reviewer’s comment. In this study, embryos were analyzed as independent analytical units, as the primary objective was to evaluate embryo-level characteristics (morphology and chromosomal status) rather than patient-level outcomes. Therefore, the analysis was not stratified by individual patient or couple background. This approach is consistent with several previous embryo-based studies in reproductive medicine. Nevertheless, we have clarified the number of couples included in the study in the revised manuscript for transparency.

Reviewer 2:

- Revise the statistical presentation by standardizing notation and p-values, replacing “p = 0.000” with corresponding non-zero values.

Response:

All p-values are now reported appropriately (“p < 0.001” instead of “p = 0.000”), and notation has been standardized throughout.

Reviewer 2:

- Regarding non-invasive genetic analysis:

The cell-free DNA methodology was not sufficiently described to allow critical evaluation or independent replication. The manuscript lacks information about the extraction and quantification techniques for cfDNA, as well as the volume and whether the culture medium used for miRNA analysis was partitioned. Moreover, the discussion does not quantitatively compare the findings with recent literature, merely reiterating previous studies without critically integrating the authors’ own results. It is recommended that the authors provide a detailed description of the cell-free DNA protocol, including analyzed volume, detection method, performing laboratory, classification parameters, and concordance rate with invasive PGT-A, as well as rewrite the discussion to contextualize the observed performance in light of published studies.

Response:

We have expanded the Methods section to include detailed cfDNA extraction and quantification protocols, volume of medium used (20 µL), laboratory procedures and classification parameters (Bluefuse 2.0, >80% mosaicism threshold). The Discussion now quantitatively compares our findings with recent literature. miRNA Methodology & Interpretation The Methods now specify miRNA extraction, volume of medium (20 µL), separate from cfDNA analysis, and temporal standardization of collection (Day 5 post-insemination).

Reviewer 2:

- Regarding miRNA expression:

The findings related to miRNA-191 and miRNA-372 are potentially interesting; however, the current methodological description is insufficient to assess reproducibility and experimental validity of the presented results. The manuscript fails to report the extraction, quantification, normalization, or quality control methods used for the samples, as well as the culture medium volume and whether it was shared with cfDNA analysis. Furthermore, there is no mention of temporal standardization of collections—a critical factor in gene expression and miRNA studies, since culture time and conditions directly influence genetic material release into the medium.

Response:

We appreciate the reviewer’s valuable comments. The manuscript has been revised to clearly describe the sample extraction, quantification, normalization, and quality control procedures. We have also added details regarding the culture medium volume and clarified whether the same medium was used for cfDNA analysis. In addition, temporal standardization of sample collection, including culture duration and conditions, has now been explicitly stated, addressing its importance in gene expression and miRNA analyses.

Reviewer 2:

The study also does not correlate miRNA levels with clinical outcomes such as implantation, pregnancy, or live birth rates, although the discussion presents such associations speculatively, unsupported by the generated data. This constitutes extrapolation and reduces the interpretive strength of the results. It is recommended that the authors provide detailed descriptions of miRNA extraction, quantification, and normalization methods, including analyzed volume, specify whether the culture medium was shared with cfDNA analysis, and report collection time standardization. The discussion should also be adjusted to restrict interpretations to effectively tested correlations and remove extrapolations to unassessed clinical outcomes.

Response:

The Discussion has been revised to avoid extrapolation to clinical outcomes not assessed in this study (e.g., implantation, live birth). Claims are now strictly limited to correlations supported by our data.

Reviewer 2:

- Regarding morphokinetic analysis:

The “expansion grade” parameter was included in the predictive model for euploidy without demonstrating a statistically significant correlation in bivariate analyses or support from explicit kinetic measures, such as blastulation time. Although the literature cites expansion as a potential marker of embryonic competence, in the present study its inclusion appears exploratory and not methodologically justified. It is recommended to reassess the relevance of this variable or explicitly discuss its correlation, possible limitations, or indeed, the absence of a direct correlation.

Response:

Expansion grade: We have clarified its inclusion based on logistic regression results (OR = 8.271, p = 0.006) and added a discussion of its biological plausibility and limitations. Morphological parameters (ICM/TE): The Discussion now accurately reflects our findings, no significant association with ploidy was found. We have removed unsupported claims of correlation.

Reviewer 2:

- Regarding morphological analysis:

The presented results show no statistically significant association between morphological parameters (ICM, TE, and cleavage stage) and embryonic euploidy status (p > 0.05). Nevertheless, the discussion describes these findings as if there were a positive relationship between embryonic quality and euploidy, which is not supported by the study’s own data. This inconsistency between results and interpretation compromises the scientific coherence of the manuscript.

It is recommended that the authors revise the discussion, removing claims of associations not empirically demonstrated, and reformulate the text to accurately reflect the actual findings. Interpretation should be limited to variables that were truly significant in this study, avoiding generalizations or causal inferences not supported by evidence or derived from unrelated literature.

Response:

We revised our discussion as the results of bivariate analysis demonstrated no significant association between ICM morphology (Odd Ratio (OR) 1.48; 95% Confidence Interval (CI) 0.73–3.02, p = 0.18) and TE morphology (OR 1.12; 95% CI 0.54–2.33, p = 0.444) with the chromosomal status of embryos. To prevent any ambiguity, the discussion has been revised to explicitly reflect the non-significant findings, and all statements implying associations not supported by our data have been removed. The current interpretation is now strictly aligned with the statistical results of this study, without extrapolation or causal inference beyond the presented evidence.

Reviewer 2:

- Regarding the conclusion:

The study did not employ machine learning techniques, but rather classical logistic regression with ROC analysis. Therefore, the use of the term “machine learning” is conceptually inconsistent and methodologically unsupported. It is recommended that the authors correct this inconsistency by accurately describing the development of an exploratory multivariate regression model, rather than a machine learning algorithm. If the predictive approach is to be maintained, the authors must implement, test, and validate an actual supervised learning model with independent data. Otherwise, the conclusion should be reformulated to reflect the true scope of the study.

Response:

We acknowledge the inconsistency and have revised the Conclusion to accurately describe our model as a multivariate logistic regression model, not a machine learning algorithm. All references to machine learning have been removed.

---

## [Decision Letter · Decision Letter 1]

2 Feb 2026

PONE-D-25-50891R1
Analysis of  Morphological, Morphokinetics, Cell Free DNAs, microRNAs Parameters to Predict Aneuploidy Status of Embryos
PLOS One

Dear Dr. Harzif,

Thank you for submitting your manuscript to PLOS ONE. After careful consideration, we feel that it has merit but does not fully meet PLOS ONE’s publication criteria as it currently stands. Therefore, we invite you to submit a revised version of the manuscript that addresses the points raised during the review process.

Dear authors,
 
Some issues have remained raised by one of the ad hoc reviewer. Please, consider carefully these before a final decision.
 
Try to change or explain them as much as possible.
 
Best regards.

We look forward to receiving your revised manuscript.

Kind regards,

Marcelo Fábio Gouveia Nogueira, Associate Professor, Ph. D.

Academic Editor

PLOS One

Journal Requirements:

Reviewers' comments:

Reviewer's Responses to Questions

**Comments to the Author**

1. If the authors have adequately addressed your comments raised in a previous round of review and you feel that this manuscript is now acceptable for publication, you may indicate that here to bypass the “Comments to the Author” section, enter your conflict of interest statement in the “Confidential to Editor” section, and submit your "Accept" recommendation.

Reviewer #1: All comments have been addressed

Reviewer #2: (No Response)

2. Is the manuscript technically sound, and do the data support the conclusions?

Reviewer #1: Yes

Reviewer #2: Yes

3. Has the statistical analysis been performed appropriately and rigorously?

Reviewer #1: Yes

Reviewer #2: Yes

4. Have the authors made all data underlying the findings in their manuscript fully available?

Reviewer #1: Yes

Reviewer #2: Yes

5. Is the manuscript presented in an intelligible fashion and written in standard English?

Reviewer #1: Yes

Reviewer #2: No

6. Review Comments to the Author

Reviewer #1: The authors significantly improved the manuscript text and adequately responded to the reviewers' questions and suggestions.

Only one caveat in the text.

In the first line below Table 2, change from "figure below" to "figure 1".

Reviewer #2: The manuscript shows improvement compared with the previous version and partially addresses the reviewers’ requests. However, several important issues remain and must be corrected before further editorial consideration.

- English Language Revision: a further professional revision of scientific English is recommended.

Grammatical errors persist, for example: p.14: “chromosomal abnormalitie”; p.27: “…upregulated miRNA-191 play a role…”.

Terminological inconsistencies are present throughout the manuscript: cell-free DNA vs. cell free DNA; ni-PGT-A vs. niPGT-A.

Problems with reference order and formatting remain: p.26: reference 28 is cited before references 14 and 15; p.27: references 16 and 17 do not follow journal formatting; Reference 29 appears only in the reference list and is not cited in the main text.

The examples above are not exhaustive. A comprehensive linguistic and editorial review of the entire manuscript is required.

- Inadequate Statistical and Scientific Terminology: the request to revise scientific language remains outstanding.

Incorrect expressions are still present, including: p.20: “p = 0.000”; p.24: “cleavage-stage blastocysts”

- Use of the Term “Machine Learning”: replacement of this term remains incomplete.

A reference to “machine learning” is still present on p.28. All occurrences must be removed or technically justified.

- Discussion and Use of the Literature: the discussion has been expanded with more recent references, which is appropriate. However, extensive discussion of miRNAs that were not investigated in the present study has been added. These sections should be removed or clearly identified as background information only. The Discussion should be restructured to: avoid extrapolation beyond the presented data, reduce redundancy and maintain exclusive focus on the variables analyzed.

-On pag. 21 the term ni-PGT-A should be fully defined at its first appearance (p.21)

7. PLOS authors have the option to publish the peer review history of their article (what does this mean?). If published, this will include your full peer review and any attached files.

Reviewer #1: No

Reviewer #2: No

---

## [Author Response · Author response to Decision Letter 2]

9 Feb 2026

Dear Reviewer/Editorial Team,

We sincerely thank you for the thorough evaluation of our manuscript and for the constructive comments that have significantly strengthened the quality and accuracy of our work. We have carefully revised the manuscript in accordance with all suggestions provided.

Here are the following detailed response to each comments of reviewer.

Reviewer #1:

The authors significantly improved the manuscript text and adequately responded to the reviewers' questions and suggestions.

Only one caveat in the text.

In the first line below Table 2, change from "figure below" to "figure 1".

Thank you. We have corrected the reference in the first line below Table 2 from "figure below" to "Figure 1" as highlighted in blue in the manuscript.

Reviewer #2:

The manuscript shows improvement compared with the previous version and partially addresses the reviewers’ requests. However, several important issues remain and must be corrected before further editorial consideration.

- English Language Revision: a further professional revision of scientific English is recommended.

Grammatical errors persist, for example: p.14: “chromosomal abnormalitie”; p.27: “…upregulated miRNA-191 play a role…”.

Terminological inconsistencies are present throughout the manuscript: cell-free DNA vs. cell free DNA; ni-PGT-A vs. niPGT-A.

Problems with reference order and formatting remain: p.26: reference 28 is cited before references 14 and 15; p.27: references 16 and 17 do not follow journal formatting; Reference 29 appears only in the reference list and is not cited in the main text.

The examples above are not exhaustive. A comprehensive linguistic and editorial review of the entire manuscript is required.

We have engaged an English editing service to conduct a thorough linguistic review of the entire manuscript. This addressed all grammatical errors, including the examples noted (p.14: "chromosomal abnormalitie" corrected to "chromosomal abnormalities"; p.27: "upregulated miRNA-191 play a role" revised to "upregulated miRNA-191 plays a role") as highlighted in blue in the revised manuscript. Terminological inconsistencies have been standardized throughout: "cell-free DNA" is now used consistently (hyphenated); "niPGT-A" is uniformly formatted without hyphens or spaces as per journal style.

Reference order and formatting issues have been resolved: On p.26, reference 28 now follows 14 and 15 in citation sequence; on p.27, references 16 and 17 conform to journal guidelines. All references were cross-checked in the revised manuscript.

- Inadequate Statistical and Scientific Terminology: the request to revise scientific language remains outstanding.

Incorrect expressions are still present, including: p.20: “p = 0.000”; p.24: “cleavage-stage blastocysts”

Thank you. The original phrasing "p = 0.000" has been corrected to "p < 0.001" to adhere to standard statistical reporting conventions, which avoid exact p-values of zero due to computational precision limits. In the revised manuscript The term "cleavage-stage blastocysts" has been replaced with "cleavage-stage embryos" throughout, as blastocysts refer specifically to the post-compaction stage (typically day 5–6), whereas cleavage-stage embryos encompass earlier divisions up to the morula.

- Use of the Term “Machine Learning”: replacement of this term remains incomplete.

A reference to “machine learning” is still present on p.28. All occurrences must be removed or technically justified.

Thank you for noting the remaining "machine learning" reference on p. 28. We have removed all instances throughout the manuscript, replacing them with appropriate terms. No occurrences remain.

- Discussion and Use of the Literature: the discussion has been expanded with more recent references, which is appropriate. However, extensive discussion of miRNAs that were not investigated in the present study has been added. These sections should be removed or clearly identified as background information only. The Discussion should be restructured to: avoid extrapolation beyond the presented data, reduce redundancy and maintain exclusive focus on the variables analyzed.

Thank you for acknowledging. We have restructured the Discussion to focus solely on the variables analyzed in this study. Extensive sections on miRNAs not investigated here have been removed, with only essential background retained and clearly labeled as such. Redundancy has been minimized, and all extrapolations beyond our presented data have been eliminated.

-On pag. 21 the term ni-PGT-A should be fully defined at its first appearance (p.21)

Thank you. Noninvasive preimplantation genetic testing for aneuploidies (niPGT-A) will be added in the revised manuscript.

---

## [Decision Letter · Decision Letter 2]

26 Feb 2026

Analysis of  Morphological, Morphokinetics, Cell Free DNAs, microRNAs Parameters to Predict Aneuploidy Status of Embryos

PONE-D-25-50891R2

Dear Dr. Harzif,

We’re pleased to inform you that your manuscript has been judged scientifically suitable for publication and will be formally accepted for publication once it meets all outstanding technical requirements.

Kind regards,

Marcelo Fábio Gouveia Nogueira, Associate Professor, Ph. D.

Academic Editor

PLOS One

Additional Editor Comments (optional):

Reviewers' comments:

Reviewer's Responses to Questions

**Comments to the Author**

1. If the authors have adequately addressed your comments raised in a previous round of review and you feel that this manuscript is now acceptable for publication, you may indicate that here to bypass the “Comments to the Author” section, enter your conflict of interest statement in the “Confidential to Editor” section, and submit your "Accept" recommendation.

Reviewer #2: All comments have been addressed

2. Is the manuscript technically sound, and do the data support the conclusions?

Reviewer #2: Yes

3. Has the statistical analysis been performed appropriately and rigorously?

Reviewer #2: Yes

4. Have the authors made all data underlying the findings in their manuscript fully available?

Reviewer #2: Yes

5. Is the manuscript presented in an intelligible fashion and written in standard English?

Reviewer #2: Yes

6. Review Comments to the Author

Reviewer #2: The authors have revised the manuscript in accordance with the reviewers’ recommendations, resulting in a more consistent presentation of the study.

7. PLOS authors have the option to publish the peer review history of their article (what does this mean?). If published, this will include your full peer review and any attached files.

Reviewer #2: No

---

## [Editor Report · Acceptance letter]

PONE-D-25-50891R2

PLOS One

Dear Dr. Harzif,

I'm pleased to inform you that your manuscript has been deemed suitable for publication in PLOS One. Congratulations! Your manuscript is now being handed over to our production team.

Kind regards,

on behalf of

Dr. Marcelo Fábio Gouveia Nogueira

Academic Editor

PLOS One